# Preparation and Characterization of Carbon Fibers from Lyocell Precursors Grafted with Polyacrylamide via Electron-Beam Irradiation

**DOI:** 10.3390/molecules26092459

**Published:** 2021-04-23

**Authors:** Hong Gun Kim, Yong-Sun Kim, Yun-Su Kuk, Lee Ku Kwac, Sun-Ho Choi, Jihyun Park, Hye Kyoung Shin

**Affiliations:** 1Institute of Carbon Technology, Jeonju University, 303 Cheonjam-ro, Wansan-gu, Jeonju-si 55069, Jeollabuk-do, Korea; hgkim@jj.ac.kr (H.G.K.); wva223g6@naver.com (Y.-S.K.); kwack29@jj.ac.kr (L.K.K.); rokmwinning2014@gmail.com (S.-H.C.); jennai@jj.ac.kr (J.P.); 2Korea Carbon Industry Promotion Agency, 111, Biseognal-ro, Deokjin-gu, Jeonju-si 54852, Jeollabuk-do, Korea; yunsu@kcarbon.or.kr

**Keywords:** lyocell, polyacrylamide, grafting, carbon fibers, electron-beam irradiation, tensile strength

## Abstract

Carbon fibers, which act as reinforcements in many applications, are often obtained from polyacrylonitrile (PAN). However, their production is expensive and results in waste problems. Therefore, we focused on producing carbon fibers from lyocell, a cellulose-based material, and analyzed the effects of the process parameters on their mechanical properties and carbon yields. Lyocell was initially grafted with polyacrylamide (PAM) via electron-beam irradiation (EBI) and was subsequently stabilized and carbonized. Thermal analysis showed that PAM grafting increased the carbon yields to 20% at 1000 °C when compared to that of raw lyocell, which degraded completely at about 600 °C. Stabilization further increased this yield to 55%. The morphology of the produced carbon fibers was highly dependent on PAM concentration, with fibers obtained at concentrations ≤0.5 wt.% exhibiting clear, rigid, and round cross-sections with smooth surfaces, whereas fibers obtained from 2 and 4 wt.% showed peeling surfaces and attachment between individual fibers due to high viscosity of PAM. These features affected the mechanical properties of the fibers. In this study, carbon fibers of the highest tensile strength (1.39 GPa) were produced with 0.5 wt.% PAM, thereby establishing the feasibility of using EBI-induced PAM grafting on lyocell fabrics to produce high-performance carbon fibers with good yields.

## 1. Introduction

Carbon fibers are widely used as reinforcement materials in the advanced composites for application in wind-turbine blades, high-tech aerospace devices, and high-quality sporting goods [1,2,3,4,5,6]. The precursors of carbon fibers are divided into different types based on their source, i.e., polyacrylonitrile (PAN), pitch, and cellulose. Among these, PAN is the most widely used precursor (up to 98%) to manufacture high-performance carbon fibers [7,8]. However, PAN sourced from fossil fuels poses several limitations, such as the generation of large amounts of toxic gases during carbonization and waste-treatment problems after use [9,10,11,12]. In addition, PAN is expensive and its cost is dependent on the variations in crude-oil cost [13,14,15,16,17]. Therefore, an environmentally friendly and inexpensive precursor is required to develop high-performance carbon fibers. Cellulose is one of the most abundant organic materials and is the oldest known precursor for carbon fibers [18]. However, the tensile strength and yield of carbon fibers developed from this precursor tend to be low [19,20]. Despite these disadvantages, cellulose-based carbon fibers should be considered instead of PAN-based carbon fibers as they offer advantages of an abundant supply of raw materials, and economic and environmental profit. Therefore, there is much interest in developing cellulose-based carbon fibers with good mechanical properties and high yields. Lyocell, unlike rayon-based fibers, which necessitate operations such as alkalization and xanthation, is a thin eco-friendly filament material based on cellulose and is produced by a spinning method through the direct dissolution of cellulose in *N*-methylmorpholine-*N*-oxide (NMMO) or a mixture of NMMO and water. In addition, lyocell fibers exhibit higher mechanical and lower shrinkage properties than rayon-based fibers [21,22,23,24,25,26]. However, similar to rayon-based fibers, lyocell fibers require pretreatment to enhance carbon yield and mechanical properties before thermal stabilization [27,28]. Generally, rayon-based carbon fibers pretreated with flame retardants exhibit higher mechanical properties and carbon yields than those obtained by a conventional thermal stabilization process without pretreatment. Mironova et al. [29] used silylated acetylene and alkoxysilanes to enhance the carbonization efficiency of lyocell. Although this process increased the carbon yield from the lyocell matrix, the mechanical properties of the resultant carbon fibers were not analyzed. Byrne et al. [30] reported a 50% increase in carbon yields from lyocell fibers by impregnating them with chemicals containing phosphate anion, but this mechanical property has also not been reported. Ongoing and various studies for lyocell-based carbon fiber are required. Therefore, further studies are required on lyocell-based carbon fibers to improve their mechanical properties and performance.

In this study, to prepare lyocell-based carbon fibers, we used an electron beam irradiation (EBI) for grafting polyacrylamide (PAM); notably, no flame retardants were used. EBI, which is a type of ionizing radiation produced by a linear accelerator, can induce reactions such as polymerization [31,32,33], grafting [34,35], cross-linking [36,37], and scission [38,39]. A few studies were conducted in the past on EBI treatment for carbon fibers. Shin et al. [40] employed EBI treatment instead of thermal treatment in the stabilization step for producing carbon fibers from a PAN precursor. Even though the used EBI dose (500–5000 kGy) was high, the gel fraction and density observed indicated a high degree of cyclization of the EBI-stabilized PAN fibers (this value increased with an increase in the EBI dose). Differential scanning calorimetry (DSC) showed that the activation energy of EBI-stabilized PAN fibers with cyclized structure reduced with an increase in the EBI dose. In addition, to develop a simple and short stabilization process, Shin et al. [41] carried out thermal treatment after the EBI pretreatment. The tensile strength of carbon fibers obtained from stabilized PAN fibers that were thermally treated at 250 °C for 40 min after EBI pretreatment (1000 kGy) was as high as 2.3 GPa. However, it should be noted here that the EBI dose was very high. Therefore, in this study, to reduce EBI dose and to achieve lyocell-based carbon fibers with good mechanical properties at high yields, we grafted lyocell with polyacrylamide (PAM) using EBI during pretreatment. Generally, PAM is grafted on cellulose to improve the physicochemical properties of the latter. In the rest of this study, we shall discuss the effect of PAM concentration and EBI doses on the performance of the produced lyocell-based carbon fibers.

## 2. Results and Discussion

### 2.1. Flourier-Transform Infrared (FT-IR) Spectroscopy

Figure 1 shows the FT-IR spectra of lyocell grafted with PAM via EBI; these spectra can be used to derive information on the physical structure and functional groups presented in the system. The sample nomenclature used in this study is shown in Table 1. Raw lyocell exhibited an intense absorption peak corresponding to −OH groups in the wavenumber range of 3000–3500 cm^−1^; in addition, peaks corresponding to C–H stretching (2850–2950 cm^−1^), C-O stretching (1024 cm^−1^), and C–O groups in anhydrous glucose rings (1640 and 1100 cm^−1^) could also be observed. Meanwhile, the spectrum of lyocell grafted with PAM peaks could be noted at 3323, 3196, 1653 cm^−1^, and 1604 cm^−1^; the intensity of these peaks, which are attributed to the -CONH_2_ group, increased with an increase in the PAM concentration. The increase in the intensity of peaks at 1452 and 1410 cm^−1^, which represent C-N stretching, was ascribed to PAM grafting. These results are evidence of the successful grafting of lyocell with PAM using EBI [42,43,44].

### 2.2. Thermal Properties of Lyocell Fabrics Grafted with PAM via EBI of before and after Thermal Stabilization

The thermal properties of the lyocell precursor significantly influence the preparation of carbon fibers or fabrics. To evaluate the carbon yield from lyocell fabrics grafted with PAM of different concentrations at 100 kGy of EBI doses, the samples were stabilized at 300 °C for 1 h, and subsequently thermogravimetric analysis (TGA) was carried out. As shown in Figure 2a, the weight of raw lyocell decreased drastically at temperatures of 290–300 °C due to lyocell degradation; additionally, its carbon yield at ~600 °C was 0%. In Figure 2b, it can be observed that the weights of lyocell fabrics grafted with PAM via EBI treatment decreased rapidly in the range of 290–300 °C; however, their carbon yield was in the range of 10–20% depending on the PAM concentration. When compared to the lyocells grafted with PAM via EBI stabilized at 300 °C for 1 h, the carbon yield was observed to be two times higher (40% and 55%) than that of lyocell grafted with PAM via EBI before stabilization. Additionally, the weight loss in the range of 300–700 °C decreased slowly with increasing PAM concentration owing to the enhanced thermal stability and conversion of the infusible ladder structure due to the exchange of H in the amide group with the alkyl groups of PAM during the stabilization [45]. Thus, carbon fabrics, using lyocell grafted with PAM via EBI, could be produced with higher thermal stability through a simple stabilization process, resulting in higher carbon yields.

### 2.3. X-ray Diffraction and Raman Studies of Carbon Fibers Obtained from Lyocell Grafted with PAM via EBI

To study the crystalline structure of the carbon fibers obtained from lyocell fabrics grafted with PAM via EBI, the samples were analyzed by X-ray diffraction (XRD). All the carbon fibers showed diffraction patterns with two peaks at 2*θ*~22° and 43°, corresponding to the (002) and (100) planes, respectively. These peaks are attributed to crystallization at different degrees of graphitization. As shown in Figure 3a and Table 1, the full width at half-maximum (FWHM) of the carbon fibers decreased slightly upon increasing the PAM concentration owing to the development of graphitic structures. In addition, the Raman spectra shown in Figure 3b were used to determine the microstructural changes occurring in the carbon fibers obtained from lyocell fabrics grafted with PAM via EBI. The Raman spectra exhibited two characteristic peaks at 1365 cm^−1^ (D-band) and at 1600 cm^−1^ (G-band). The D-band is related to disordered or defective graphite structures, while the G-band is related to an ordered–layered graphite structure. To evaluate the degree of graphitization, the intensity ratio of these peaks (*I_G_/I_D_*) was employed. As shown in Figure 3b, although there were minimal changes in the peak positions of the D- and G- bands, as the PAM concentration increased, the *I_G_/I_D_* ratio increased slightly. These ratios were calculated to be 1.01, 1.03, 1.04, 1.05, 1.05, and 1.06 for carbon fibers obtained from L-0.05-100, L-0.1-100, L-0.5-100, L-1-100, L-2-100, and L-4-100, respectively. These changes may be owing to increased crystallization due to the grafting of PAM on lyocell.

### 2.4. Morphology of Lyocell Grafted with PAM via EBI and Subjected to Stabilization and Carbonization

Figure 4 shows the cross-sectional images of the raw lyocell, lyocell grafted with PAM via EBI, and PAM-grafted lyocell subjected to thermal stabilization and thus carbonization. Raw lyocell and lyocells grafted with PAM via EBI showed smudged morphologies owing to fiber softness. In addition, the surface of the L-0.05-100, L-0.1-100, and L-0.5-100 samples was clear and smooth, indicating good interfacial adhesion between the lyocell fibers and PAM without adhering to neighboring fibers. With a gradual increase in PAM concentration, the surfaces of the fibers were surrounded by large amounts of PAM. After stabilization, it was observed that the L-0.05-100, L-0.1-100, and L-0.5-100 samples exhibited mostly rigid, clear, and circular cross-sections with smooth surfaces; they could endure higher temperatures than the raw lyocell and lyocells grafted with PAM via EBI without stabilization. The L-1-100 and L-2-100 samples exhibited a rigid cross-section and their surfaces were peeled off; moreover, some L-4-100 fibers were attached to each other due to immobilization at high PAM concentrations (implying high viscosity). Meanwhile, after carbonization, the L-0.05-100, L-0.1-100, and L-0.5-100 carbon fibers exhibited rigid, clear, and circular cross-sections with smooth surfaces. They could be separated well into single filaments for measuring their tensile strengths. The L-1-100, L-2-100, and L-4-100 carbon fibers exhibited rigid and round cross-sections and peeling surfaces. In the case of L-4-100 carbon fibers, once again, some of the fibers were attached to each other. These morphological differences are expected to affect the mechanical properties of the carbon fibers obtained from lyocell fabrics grafted with PAM via EBI.

### 2.5. Mechanical Properties of Carbon Fibers Obtained from Lyocells Grafted with PAM via EBI

Figure 5 evaluates the tensile strength of carbon fibers isolated from carbon fabrics obtained from lyocell grafted with PAM via EBI. Carbon fibers obtained from the L-0.05-100, L-0.1-100, and L-0.5-100, and L-1-100 have rigid, clear, and circular cross-sections with smooth surfaces as evident from the SEM images. Accordingly, the tensile strength of these produced carbon fibers increased from ~0.82 to 1.39 GPa.

PAM grafting via EBI is thus a useful method to improve the mechanical properties of lyocell-based carbon fibers, even though these values are lower than those of PAN-based carbon fibers (>2.5 GPa). In addition, the tensile strength of the carbon fibers obtained from the L-2-100 and L-4-100 fibers could not be measured because of the brittleness of the produced carbon fibers as well as the attachment between carbon fibers owing to the high viscosity of PAM (as shown in the SEM images in Figure 3), and the tensile strength of carbon fibers obtained from L-1-100 was lower than that of fibers obtained from L-0.5-100 due to damage of fiber during separation of a single filament. Figure 6 shows photographs of the L-0.5-100 fabric and the carbon fabric obtained from it. Evidently, the shape of the produced carbon fabric was similar to that of the L-0.5-100 fabric and could be bent and rolled without breaking, showing its superior flexibility.

## 3. Methods and Materials

### 3.1. Materials

Lyocell fabrics were procured from Hyosung Co. (Ulsan, Korea) while PAM was purchased from Sigma-Aldrich (St. Louis, MO, USA).

### 3.2. Grafting of Lyocell Fabrics with PAM via EBI and Their Thermal Stabilization and Carbonization

As illustrated in Figure 7, lyocell fabrics (10 × 10 cm^2^) were immersed in a solution of PAM (0.05, 0.1, 0.5, 1, 2, and 4 wt.%) dissolved in water, and they were irradiated with EBI of various dosages (100, 200, and 300 kGy); however, in our analysis, we observed that the effect of PAM concentration was higher than that of the EBI dose on the properties of carbon fibers. Therefore, we selected the lowest EBI dose of 100 kGy for grafting PAM onto lyocell fabrics. EBI treatment was carried out at an accelerating voltage of 1.14 MeV with a beam current of 7.6 mA, irradiation width of 110 cm, and dose rate of 6.67 kGy/s in ambient air atmosphere at room temperature. The treated lyocell fabrics were freeze-dried and thermally stabilized at 300 °C for 1 h. Subsequently, they were carbonized at 900 °C (heating rate of 10 °C/min) in a tubular furnace in a N_2_ (99.999%) atmosphere. The lyocell fabric samples, grafted with different concentrations of PAM at 100 kGy of EBI dose, were labeled as L-0.05-100, L-0.1-100, L-0.5-100, L-1-100, and L-2-100. In addition, Table 2 shows the amounts of PAM grafting on the lyocell fabrics. These values were obtained by subtracting the weight of raw lyocell fabric from the lyocell fabric grafted with PAM.

### 3.3. Characterization

FT-IR spectroscopy was conducted using a Nicolet iS50 (Thermo Fisher Scientific, Waltham, MA, U.S.) instrument equipped with an attenuated total reflectance (ATR) accessory. The spectra were acquired in the wavenumber range of 4000–400 cm^−1^ at a resolution of 4 cm^−1^ over 32 scans.

The thermal stability of lyocell fabrics grafted with PAM via EBI and their thermally stabilized counterparts was evaluated by TGA (TA, SDT Q600, New Castle, DE, U.S.). In this test, samples (>10 mg) were heated in the range of 30–1000 °C at a heating rate of 10 °C/min in a pure N_2_ atmosphere. The crystalline structures of carbon fabrics obtained from lyocell fabrics grafted with PAM via EBI were evaluated by XRD (RIGAKU, D/MAX-2500, Tokyo, Japan) at an operating voltage and current of 40 kV and 30 mA, respectively, with CuKα radiation. Raman spectra were obtained on an ARAMIS instrument (Horiba Jobin Yvon, Tokyo, Japan) with a 514 nm laser in the spectral range of 500–3500 cm^−1^.

The tensile strengths of filaments isolated from carbon fabrics obtained from lyocell fabrics grafted with PAM via EBI were analyzed on a Favigraph (Textechno, Mönchengladbach, Germany) instrument equipped with a load cell of 100 cN. Tensile tests were conducted at least 20 times per sample, and the average results were reported.

## 4. Conclusions

In this study, to develop high-performance and sustainable carbon fibers at high yields, we grafted lyocell fabrics with PAM using EBI and subsequently subjected them to thermal stabilization and carbonization. TGA showed that PAM grafting on lyocell fabrics using EBI increased carbon yield from 10% to 20% at 1000 °C when compared to raw lyocell, which degraded completely at 600 °C. In addition, by stabilization, the carbon yield of the fabrics could be increased to values as high as 55%. Furthermore, SEM indicated that the morphologies of the prepared carbon fibers were highly dependent on PAM concentration, while carbon fibers obtained from the L-0.05-100, L-0.1-100, and L-0.5-100 exhibited clear, rigid, and round cross-sections with smooth surfaces and could be easily separated into single filaments. The carbon fabrics obtained from the L-1-100, L-2-100, and L-4-100 exhibited rigid and round cross-sections with peeling surfaces due to the high viscosity of PAM. Furthermore, at a PAM concentration of 4 wt.%, some fibers were attached to each other as well. All these factors influenced the mechanical properties of the resultant carbon fibers. Upon increasing the PAM concentration from 0.05 to 0.5 wt.%, the tensile strength of the produced carbon fibers could be enhanced from 0.82 to 1.39 GPa. Thus, it is evident that PAM grafting on lyocell fabrics at a low EBI dose is highly useful for improving the mechanical properties of lyocell-based carbon fibers.

## Figures and Tables

**Figure 1 molecules-26-02459-f001:**
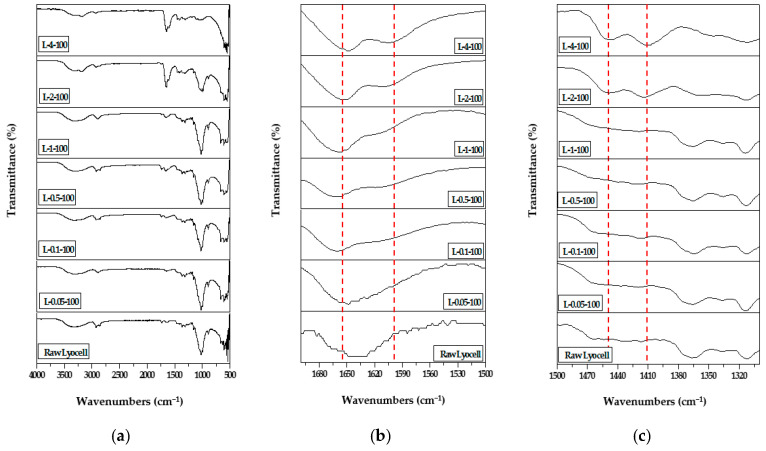
FT-IR spectra of raw lyocell fabric and lyocell fabrics (L-0.05-100, L-0.1-100, L-0.5-100, L-1-100, L-2-100, and L-4-100) grafted with different concentrations (0.05, 0.1, 0.5, 1, 2, and 4 wt.%) of PAM via 100 kGy of EBI dose in the range of (**a**) in the range of 500–4000, (**b**) 1700–1500, and (**c**) 1500–1300 cm^−1^.

**Figure 2 molecules-26-02459-f002:**
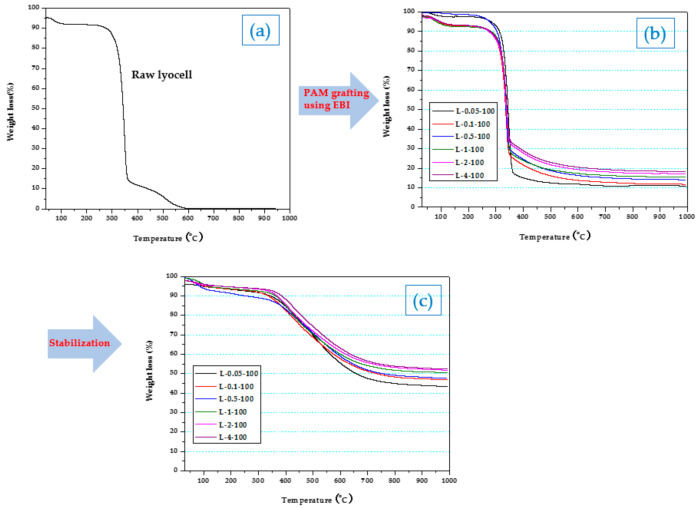
TGA analysis of (**a**) raw lyocell, (**b**) lyocell grafted with PAM via EBI before stabilization, and (**c**) lyocell grafted with PAM via EBI after thermal stabilization.

**Figure 3 molecules-26-02459-f003:**
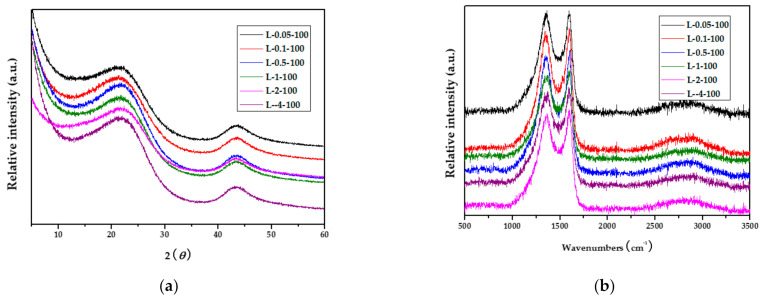
(**a**) XRD profiles and (**b**) Raman spectra of carbon fibers obtained from lyocell fabrics grafted with PAM via EBI.

**Figure 4 molecules-26-02459-f004:**
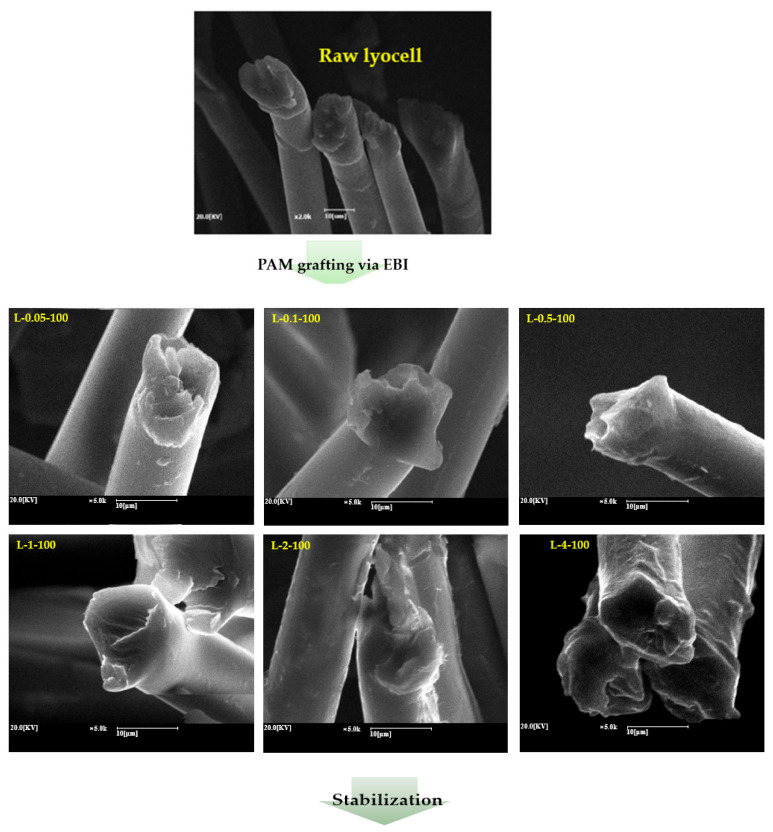
SEM images of raw lyocell, lyocell grafted with PAM of different concentrations via EBI, and PAM-grafted lyocell after stabilization and carbonization.

**Figure 5 molecules-26-02459-f005:**
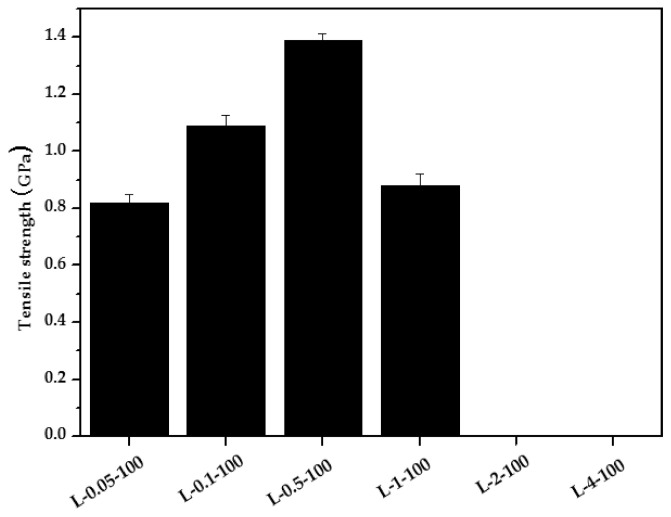
Tensile strength of carbon fibers obtained from lyocell fabrics grafted with PAM via EBI.

**Figure 6 molecules-26-02459-f006:**
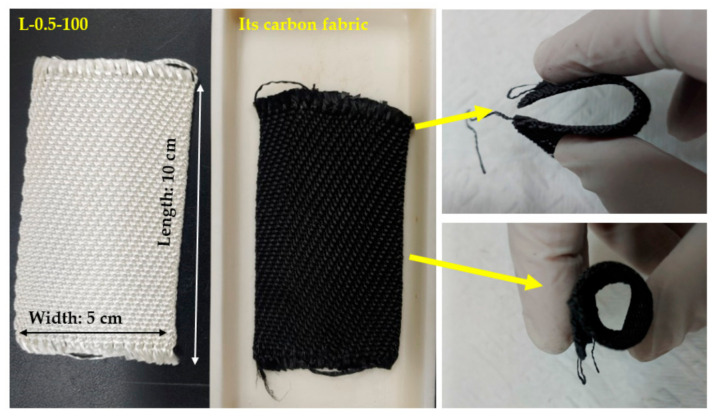
Photographs of lyocell fabric grafted with PAM via EBI and the corresponding carbon fabrice.

**Figure 7 molecules-26-02459-f007:**
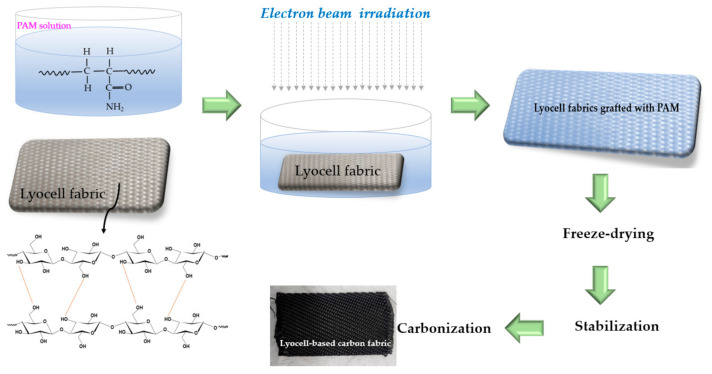
Schematic illustration of the preparation of carbon fabrics from lyocell fabrics grafted with PAM via EBI, stabilization, and carbonization.

**Table 1 molecules-26-02459-t001:** XRD data corresponding to the carbon fibers obtained from lyocell fabrics grafted with PAM via EBI.

Sample (Carbon Fibers)	Peak Center (°)	FWHM (°)	Planar Spacing (Å)	Peak Intensity (cps)
L-0.05-100	22.51	10.24	3.95	6159.21
	43.72	6.54	2.07	3220.66
L-0.1-100	22.21	10.20	4.00	7275.14
	43.46	6.67	2.08	3447.17
L-0.5-100	22.67	9.94	3.92	10,099.81
	43.33	7.05	2.08	3612.49
L-1-100	22.92	9.71	3.88	9581.71
	43.30	6.62	2.09	2734.76
L-2-100	22.80	9.50	3.90	9151.48
	43.45	6.97	2.08	2958.06
L-4-100	22.87	9.50	3.89	1130.32
	43.85	7.08	2.06	3800.02

**Table 2 molecules-26-02459-t002:** Amount of PAM grafting on lyocell fabrics.

	L-0.05-100	L-0.1-100	L-0.5-100	L-1-100	L-2-100	L-4-100
Amount of PAM grafting on lyocell fabrics (g cm^−2^)	0.0034 ± 0.00012	0.0038 ± 0.00023	0.0087 ± 0.00035	0.0112 ± 0.00050	0.0123 ± 0.00061	0.0159 ± 0.00045

## Data Availability

The data are available by corresponding author.

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
