# Peer review of "Preparation and Characterization of Carbon Fibers from Lyocell Precursors Grafted with Polyacrylamide via Electron-Beam Irradiation"

_molecules, 2021, doi:10.3390/molecules26092459_

Round 1

Reviewer 1 Report

The authors show how to make and analyse carbon fibers from cellulose based precursors. Their scheme seems to work, however, there are a few points the authors should take into account:

On all the SEM pictures scale bars are missing/are not readable.

Starting in line 193 on page 7 the authors give a short recipe on how to make even better carbon fibers. Why did you not do this? If you know how to achieve better results, go do the experiments and show them.

In some cases it was not possible to measure the tensile as the fibers could not be extracted from the fabric. Why did you not repeat the treatment with single fibers extracted befor the reactions?

Author Response

Dear reviewer

We are thankful the reviewer for their constructive comments on this manuscript.

Accordingly, the following revisions were made in the manuscript.

On all the SEM pictures scale bars are missing/are not readable.

→ Based on your comments, we have revised SEM images to improve the readability of the scale bars.

Starting in line 193 on page 7 the authors give a short recipe on how to make even better carbon fibers. Why did you not do this? If you know how to achieve better results, go do the experiments and show them.

→ In this sentence, we have referred to “Dumanli, A.F.; Windle, A.H., Carbon fibers from cellulosic precursors: a review, J. Mater. Sci. 2012, 47, 4236-4250.”. However, based on your comments, it is better to delete this sentence. In addition, we are continuously studying and doing the experiments to achieve better results. These improved results on lyocell-based carbon fibers will be published in near future.

In some cases it was not possible to measure the tensile as the fibers could not be extracted from the fabric. Why did you not repeat the treatment with single fibers extracted before the reactions?

→ We carbonized only single fibers extracted from the fabric and then tried to measure their tensile strengths. However, the obtained carbon fibers were highly brittle and still attached together as evident from the photo (L-4-100) shown below:

Therefore, we have modified “due to attachment between carbon fibers owing to the high viscosity of PAM” to “because of the brittleness of the produced carbon fibers as well as the attachment between carbon fibers owing to the high viscosity of PAM”.

Thank you again for your valuable comments and insightful suggestions.

Best regards.

Hye Kyoung Shin

Reviewer 2 Report

Molecules 1172624

Preparation and characterisation of carbon fibres...

By Hong Gun Kim et al.

Authors describe a study to prepare carbon fibres from modified cellulose fibres. Electron-beam irradiation was used to graft polyacrylamide on lyocell fibres.

The process conditions of carbon fibre production are discussed and the resulting fibres are characterised.

The material is of interest, however major revision is recommended before publication.

Major reasons:

  • The manuscript is very descriptive and there very little interpretation of findings. E.g. why thermal stabilisation leads to different amount of weight loss? Is this not a question of rate of decomposition? What is the chemical background behind this “stabilisation”.
  • The description of the preparation is incomplete: which solvent was used for PAM impregnation, how the samples were dried (if they were dried) before irradiation.
  • There is no indication about the amount of PAM fixed on the material (at least an estimate would be helpful, or N-analysis
  • Figure 7 and 8 exhibit a scheme which is questionable in the present form: which DS was achieved (here authors draw a DS of 1 in Figure 7) and 2 in Figure 8. The reaction will be governed by a surface reaction.
  • When the tensile strengths were determined, assumptions with regard to determination of area of cross section should be required.

Minor reasons:

All material was treated with the same dose, so this could be simplified both in the sample identification, and Table 1.

The use of some terms e.g. thermal stabilisation, nomenclature is questionable and should be checked.

The scale bar in Fig 4 should be readable

There are some English problems e.g. line 195

Author Response

Response to reviewer

I thank the reviewer for their very constructive comments on this manuscript.

Accordingly, the following revisions were made in the manuscript.

Major reasons:

  • The manuscript is very descriptive and there very little interpretation of findings. E.g. why thermal stabilisation leads to different amount of weight loss? Is this not a question of rate of decomposition? What is the chemical background behind this “stabilisation”.   
  •  
  • After: When compared to the lyocells grafted with PAM via EBI was stabilized at 300 °C for 1 h, the carbon yield was observed to be two times higher (40% and 55%) than that of lyocell grafted with PAM via EBI before stabilization. Additionally, the weight loss in the range of 300-700 °C decreased slowly with increasing PAM concentration owing to the enhanced thermal stability and conversion of the infusible ladder structure due to the exchange of H in the amide group with the alkyl groups of PAM during the stabilization [45].
  • Before: When compared to lyocells grafted with PAM via EBI was thermally stabilized at 300 °C for 1 h, the carbon was observed to be two times higher (40% and 55%) than that of lyocell grafted with PAM via EBI before thermal stabilization, and the weight losses in the range of 300-700 °C slowly decreased.
  • → Based on your comments, we have made the following modification in the revised manuscript:
  • The description of the preparation is incomplete: which solvent was used for PAM impregnation, how the samples were dried (if they were dried) before irradiation. 
  • → As mentioned in this manuscript, the lyocell fabrics (10×10 cm2) were immersed in PAM solutions (0.05, 0.1, 0.5, 1, 2, and 4 wt.%), and they were irradiated at 100 kGy of EBI dose. Water was used as the solvent to dissolve PAM. Accordingly, the following phrase has been included in line 213 on page 10 of the revised manuscript: “lyocell fabrics (10×10 cm2) were immersed in a solution of PAM (0.05, 0.1, 0.5, 1, 2, and 4 wt.%) dissolved in water”
  • There is no indication about the amount of PAM fixed on the material (at least an estimate would be helpful, or N-analysis
  • → We have added the amount of PAM grafting on lyocell fabrics in Table 2, as:

In addition, Table 2 shows the amounts of PAM grafting on the lyocell fabrics. These values were obtained by subtracting the weight of raw lyocell fabric from the lyocell fabric grafted with PAM.

Table2. Amount of PAM grafting on lyocell fabrics

L-0.05-100

L-0.1-100

L-0.5-100

L-1-100

L-2-100

L-4-100

Amount of PAM grafting on lyocell fabrics

(g cm-2)

0.0034±0.00012

0.0038±0.00023

0.0087±0.00035

0.0112±0.00050

0.0123±0.00061

0.0159±0.00045

  • Figure 7 and 8 exhibit a scheme which is questionable in the present form: which DS was achieved (here authors draw a DS of 1 in Figure 7) and 2 in Figure 8. The reaction will be governed by a surface reaction. 
  • → We deleted Figure 8 because it seems to overlap Figure 7 and 8 as highlighted by you.
  • When the tensile strengths were determined, assumptions with regard to determination of area of cross section should be required.

→Based on your comments, we have revised “As the PAM concentration increased from 0.05 wt.% to 0.5 wt.%, the tensile strength of the produced carbon fibers increased from ~0.82 to 1.39 GPa” to “Carbon fibers obtained from the L-0.05-100, L-0.1-100, and L-0.5-100, and L-1-100 with rigid, clear, and circular cross-sections with smooth surfaces as evident from the SEM images. Accordingly, the tensile strengths of these produced carbon fibers increased from ~0.82 to 1.39 GPa.”

Minor reasons:

All material was treated with the same dose, so this could be simplified both in the sample identification, and Table 1.

→ Based on your comments, we have deleted Table 1 and included the following simplified sentence in the revised manuscript:

The lyocell fabric samples, grafted with different concentrations of PAM at 100 kGy of EBI dose, were labeled as L-0.05-100, L-0.1-100, L-0.5-100, L-1-100, and L-2-100. In addition, Table 2 shows the amounts of PAM grafting on the lyocell fabrics.

The use of some terms e.g. thermal stabilisation, nomenclature is questionable and should be checked.

→ As per your suggestions, we have revised “thermal stabilization”to “stabilization”

The scale bar in Fig 4 should be readable

→ As per your comments, we have modified the indicated figure to improve the readability of the scale bars in the SEM images.

There are some English problems e.g. line 195

→ We have deleted line 195 to incorporate the suggestions and comments of another reviewer.

Thank you again for your valuable suggestions and comments on this manuscript

Best regards.

Hye Kyoung Shin

Round 2

Reviewer 1 Report

THanks for clarifying my questions. I think that the manuscript can now be published.

Author Response

Authors thank Reviewer's comments.

Reviewer 2 Report

The authors have revised their version accordinglyl. The only concern which is remaining is the already commentec criticism to the chemical formula present in Figure 7 which represents a chemical structure of grafted product which most probably is not existing and also is not supported by the findings. There would be no need to keep this formula.

Author Response

We are thankful the reviewer for their constructive comments on this manuscript.

Accordingly, the following revisions were made in the manuscript.

Comment: Figure 7. Schematic illustration of the preparation of carbon fabrics from lyocell fabrics grafted with PAM via EBI, thermal stabilization, and carbonization. The chemical structures represent most plausible structures but not have been verified.

Response:  For the chemical structure, we have referred to “Yuan, N.; Xu, L.; Ye, H.; Zhao, J.; Liu, Z.; Rong, J. Superior hybrid hydrogels of polyacrylamide enhanced by bacterial cellulose nanofiber clusters. Mater. Sci. Eng C 2016, 67, 221-230.: Liu, H.; Yang, X.; Zhang, Y.; Zhu, H.; Yao, Y. Flocculation characteristics of polyacrylamide grafted cellulose from phyllostachys heterocyla: An efficient and eco-friendly flocculant. Water Research 2014, 165-171.: Hong, K.H.; Liu, N.; Sun, G. UV-induced graft polymerization of acrylamide on cellulose by using immobilized benzophenone as a photo-initiator. Eur. Polym. J. 2009, 45, 2443-2449.”

 However, based on your comments, we revised the parts of PAM-g-lyocell fabric in Figure 7.